# Pharmacists’ Knowledge, Attitude, and Practice of Medication Therapy Management: A Systematic Review

**DOI:** 10.3390/healthcare10122513

**Published:** 2022-12-12

**Authors:** Farida Rendrayani, Sofa Dewi Alfian, Wawan Wahyudin, Irma Melyani Puspitasari

**Affiliations:** 1Department of Pharmacology and Clinical Pharmacy, Faculty of Pharmacy, Universitas Padjadjaran, Sumedang 45363, West Java, Indonesia; 2Center of Excellence for Pharmaceutical Care Innovation, Universitas Padjadjaran, Sumedang 45363, West Java, Indonesia; 3Ciloto Health Training Centre, Ministry of Health Republic of Indonesia, Cianjur 43253, West Java, Indonesia

**Keywords:** pharmacists, professional practices, pharmaceutical services, drug therapy management

## Abstract

Understanding pharmacists’ knowledge, attitudes, and practices (KAP) and their perceptions of challenges, barriers, and facilitators towards medication therapy management (MTM) provision are vital in informing the design and implementation of successful service delivery. Thus, this review examined pharmacists’ knowledge, attitudes, and practices, and their perceived challenges, barriers, and facilitators to MTM services provision, globally. A systematic search was conducted on 1–31 August 2022 to identify relevant studies on PubMed and EBSCO, supplemented with a bibliographic and a particular hand search. We focused on original research in quantitative survey form with the key concepts of “medication therapy management”, “pharmacists”, and “knowledge, attitude, practice”. We assessed the reporting quality using the Checklist for Reporting of Survey Studies (CROSS). Results are reported narratively and according to the Preferred Reporting Items for Systematic Reviews and Meta-Analyses (PRISMA) guideline. From 237 articles identified, 17 studies met the eligibility criteria. Of the included studies, five reported that pharmacists had a considerable level of knowledge, seven suggested a positive attitude among pharmacists, and five revealed that pharmacists had been practicing some MTM elements. Factors associated with pharmacists’ KAP may include age, educational degree, additional qualification, income, years of practice, practice setting, and experience in patient care service. The challenges toward MTM provision were related to the pharmacist-patient and pharmacist-physician relationship. Insufficient time, staff, compensation, and training were the barriers, while patients’ willingness to participate and educational background were the facilitators of MTM provision. These findings of studies on KAP could help develop an MTM program and design an intervention to improve program effectiveness. Further research focusing on other quantitative and qualitative studies of KAP is needed to obtain a comprehensive approach to MTM provision.

## 1. Introduction

Medication therapy management (MTM) is the current patient-oriented service driven by the “pharmaceutical care” philosophy [1]. Through the *Medicare Modernization Act* passage in 2003, the government of the United States introduced MTM to address the issues of increasing adverse drug events and medication costs [1]. A year later, a consensus of pharmacy professionals developed the definition of MTM. The establishment of MTM core elements in 2005 (version 1.0) and 2008 (version 2.0) serves as a model framework to implement effective MTM [2,3]. The core elements consist of a medication therapy review (MTR), a personal medication record (PMR), a medication action plan (MAP), intervention and or referral, and documentation and follow-up [2]. Application of the model is critical for improving patient outcomes and ensuring continuity of care [1,2]. Some countries have also introduced similar medication management practices, such as Medication Use Review (MUR) and Medicines Optimization in Care Homes (MOCH) in the UK, and Residential Medication Management Review (RMMR) and Quality Use of Medicine (QUM) in Australia [3,4]. Nevertheless, many countries, including Brazil, China, Ethiopia, Germany, India, Indonesia, Saudi Arabia, and Thailand, have adopted the USA MTM model for daily practice [5,6,7,8,9,10,11,12].

Pharmacists are the potential providers of MTM services because of their medication knowledge and patient access [1]. The role of pharmacists in MTM delivery has led to improved medication adherence and clinical outcomes and reduced hospitalization numbers [7,10] and medication costs [7,13,14]. Nevertheless, the success of the MTM program depends on several factors, including pharmacists themselves. As reported by Adeoye et al., (2018) [15], pharmacists’ attitude affects the rates of MTM completion. According to Elayeh et al., (2017) [16], a lack of therapeutic knowledge and clinical problem-solving skills hinders the implementation of patient-centered services. Hence, an assessment of pharmacists’ knowledge is critical for anticipating such challenges during MTM provision and comprises an essential component of strategic planning for program sustainability [17].

Knowledge, attitude, and practice (KAP) surveys could provide information regarding the needs, issues, and challenges of developing effective public health programs and postintervention data for program evaluation [18]. Several studies have used KAP surveys both before and after MTM implementation [19,20,21,22,23,24].Each study uncovers findings that could be useful for program planning or further research. A previous review examined survey articles regarding MTM programs in 2012 that investigated the perspectives of healthcare professionals, students, and patients in the US only [25]. It reported that pharmacists claimed to have sufficient knowledge and experience to provide MTM services, and pharmacy students, physicians, and that patients have positive attitudes toward the provision of MTM [25]. The study further reported that barriers to MTM provision include reimbursement, time, workload, staff, and access to patients’ records [25].

However, no studies specifically studied pharmacists around the world. This has become increasingly essential, especially considering the recent worldwide adoption of the MTM program. Various settings in the studies enable a thorough grasp of the context issue. It is very critical to help achieve and maintain effective MTM service delivery. Thus, this review examined pharmacists’ knowledge, attitudes, and practices and their perceived challenges, barriers, and facilitators to MTM services provision globally. We focused on obtaining a comprehensive understanding of the provision of MTM services, particularly regarding the challenges, barriers, and facilitators reported by pharmacists.

## 2. Materials and Methods

We followed the recent Preferred Reporting Items for Systematic Reviews and Meta-Analyses (PRISMA) 2020 guideline to report this review [26]. The PRISMA checklists of this study are presented in Appendix A.

### 2.1. Search Strategy

We performed a systematic search on 1–31 August 2022 to identify relevant studies on PubMed and EBSCO, using predetermined search strategies. The search strategies were checked by an information specialist based on the Peer Review of Electronic Search Strategies (PRESS) guideline [27]. The search strategies contained in the PRESS guideline are provided in Appendix A. We set the search terms based on the key concepts (medication therapy management, pharmacist, and knowledge, attitude, practice) and ensured them by testing several relevant studies. We used Boolean operators such as “OR” to broaden the search of each concept and “AND” to specify relevant results. On PubMed, the publication date was set from January 2008 until August 2022 to avoid irrelevant results concerning medication management because “medication therapy management” was introduced as a Medical Subject Headings (MeSH) term in 2008. We did not restrict the publication date in EBSCO. A hand search was conducted by bibliographic “snowballing” and a particular search was used to supplement the database search. We included the relevant studies published before 2008 in the hand search.

### 2.2. Study Selection

All database search results were exported to Zotero version 6.0.13 (Corporation for Digital Scholarship, Vienna, VA, USA) and checked for duplicates. One researcher (FR) independently performed the screening process, starting with the title and abstract. The inclusion criteria were original research, quantitative surveys, studies that assessed pharmacists’ knowledge, attitude, or practice toward MTM provision, and published in English. We excluded abstracts from conference proceedings, case reports, commentaries, editorials, or study protocols. The full texts of potentially eligible articles were retrieved and reviewed by FR and then independently verified by SDA and IMP.

### 2.3. Review Process

Data from the included studies were manually extracted into Microsoft^®^ Excel^®^ 2019 MSO version 2210 (Microsoft Corporation, Redmond, WA, USA) using a predefined format. Information concerning authors, title, publication year, journal, study design, survey instruments, population of interest, country, sample size, response rate, respondents’ age, educational degree, years of practice, assessment method, instrument validation, main findings, author’s interpretation, and funding were independently extracted by FR. SDA and IMP independently verified the extracted data. We used descriptive statistics to express the proportions of studies with particular characteristics. The review results were presented narratively, which involved a discussion by all researchers to ensure the accuracy of the findings.

### 2.4. Quality of Reporting

We assessed the reporting quality of each included study using the Consensus-based Checklist for Reporting of Survey Studies (CROSS) [28]. The checklist consists of nineteen items, divided into six main categories (title and abstract, introduction, methods, results, discussion, and other sections). We calculated a percentage score with the underlying assumption that all criteria were weighted equally after excluding the criteria that were not applicable. Studies were assigned 1 point for reporting the item, 0.5 for partially reporting, and 0 for not reporting. Studies were categorized as ‘high quality’ if they met at least 75% of these standards, ‘moderate’ if they met between 50% and 75% of relevant standards, and ‘low’ if less than 50%. Since this assessment only measures the quality of reporting, the fact of unreported items does not imply poor study quality.

## 3. Results

### 3.1. Study Characteristics

We identified 184 and 47 records from the database search on PubMed and EBSCO, respectively. After removing five duplicates, we screened 226 records and selected 14 articles for full-text screening. We further excluded two articles because they did not specifically discuss MTM [29,30], and another article because pharmacists’ responses could not be distinguished from those of other responders [31]. The hand search yielded six articles. As a result, there were 17 articles for the final review [32,33,34,35,36,37,38,39,40,41,42,43,44,45,46,47,48]. The PRISMA flowchart for the study selection process is depicted in Figure 1.

Table 1 summarizes the characteristics of the included studies. The majority of studies focused on pharmacists’ practice regarding MTM services. Recently, online surveys have become preferable over hand-delivered, mailed or telephoned surveys. Although most studies were conducted in the USA, there were also studies conducted in Jordan, Lebanon, Malaysia, Nigeria, and Saudi Arabia. Only two studies were national surveys that included more than 500 participants.

### 3.2. Pharmacists’ Knowledge

We reviewed seven articles on pharmacists’ knowledge about MTM in Saudi Arabia, Jordan, Nigeria, Malaysia, Lebanon, and the USA [32,33,35,36,37,38,39]. Five studies conducted in Saudi Arabia, Jordan, Nigeria, Malaysia, and Lebanon reported that pharmacists had a considerable knowledge level regarding MTM. Two studies conducted in the USA reported that training significantly improved pharmacists’ knowledge, especially in disease-specific MTM. The detailed information about the studies on knowledge is presented in Table 2.

Two recent studies on knowledge of MTM were online surveys conducted in Saudi Arabia and Jordan [32,33]. The study in Saudi Arabia used six true/false questions regarding the definition, core elements, goals, beneficiaries, roles, and providers of MTM [32]. Jarab et al., (2022) in Jordan used two true/false questions about the benefits and role of MTM [33]. They also added two multiple-choice questions about the core elements and goals of MTM where participants could select more than one answer (first question: five answers, second question: three answers) [33]. The study reported that pharmacists in Saudi Arabia had good knowledge about MTM (average score was 4.30 ± 1.00) [32], while community pharmacists in Jordan had a moderate level of knowledge (median score was six; the scores ranged from four to seven) [33].

A direct, self-administered survey in Malaysia also used multiple-choice and true/false questions [36]. The questionnaire consisted of 10 questions about MTM and the Medication Therapy Adherence Clinic (MTAC). The study classified knowledge scores into high (10–8 scores), moderate (7–6 scores), and low (5–0 scores) levels. It reported that 97.5% and 2.5% of hospital pharmacists had high and moderate knowledge levels, respectively [36]. Other direct surveys in Nigeria and Lebanon evaluated the pharmacists’ knowledge based on the response to the yes/no question [35,37]. Akonoghrere et al., (2020) in Nigeria reported that 94% of pharmacists were aware of MTM, 100% were aware of pharmaceutical care, and 98% admitted the similarities between MTM and pharmaceutical care [35]. The authors suggested that most pharmacists had good knowledge of MTM [35]. The study in Lebanon discovered that 46% of pharmacists were familiar with the MTM concept, and >60% of pharmacists who would provide the service were knowledgeable about the process of avoiding drug-related problems [37]. Thus, they were considered to have adequate knowledge of MTM services [37].

The research on knowledge conducted in the USA consisted of longitudinal studies evaluating the effectiveness of disease-specific MTM training. Pretraining and post training assessments were used to determine the effectiveness of training in improving participants’ knowledge. The assessment used multiple-choice questions related to specific disease care. The study of Brown et al., (2018) on hypertension MTM adjusted the answer correctness with participants’ confidence to minimize guessing bias [38]. The average scores of correct answers (%) adjusted by participants’ confidence were 71.0 (pretraining) and 82.4 (post training) (*p* < 0.05) [38]. Another study conducted by Battaglia et al., (2012) on diabetes MTM reported the average correct answer scores of 11 questions as 7.1 (pretraining) and 8.0 (post training) (*p* < 0.01) [39]. Both of these studies found that training could significantly improve the knowledge of disease-specific MTM.

Among all studies, only four performed content validation, and five pre-tested the questionnaires. Here, content validity refers to the degree to which items on a measure assess the same content, as determined by experts [49,50]. Regarding the reliability of the questionnaire, only the study conducted in Malaysia reported internal consistency through a Cronbach’s alpha of 0.326 [36].

All of the studies on knowledge evaluated participants’ sociodemographic characteristics, including at least one of the following parameters: age, educational degree, and years of practice. Two studies discovered that age, educational degree, income, and pharmacy practice setting at the hospital were associated with knowledge level [33,36].

### 3.3. Pharmacists Attitude

Table 3 details the eleven articles on attitudes toward the provision of MTM services. Seven studies reported that pharmacists had a positive attitude [32,33,34,35,36,37,43], one study mentioned that pharmacists had a slightly positive attitude [40], and another study stated that pharmacists were willing to provide MTM [41]. Moreover, one study reported that training did not improve pharmacists’ attitudes [39], whereas another study indicated that contracting might be associated with pharmacists’ attitudes toward the MTM benefit to patient outcomes [42].

Alshehri et al., (2022) in Saudi Arabia used the degree of importance placed on the provision of MTM services to define pharmacists’ attitudes [32]. The assessment used four items on a seven-point Likert scale (1 = harmful and 7 = beneficial; 1 = bad and 7 = good; 1 = unpleasant for me and 7 = pleasant for me; 1 = worthless and 7 = useful). The average score was 6.15 ± 1.12, which indicated a positive attitude [32].

Three studies conducted in Jordan, Malaysia, and the USA used a 5-point Likert scale and reported the average agreement score [33,36,43]. Jarab et al., (2022) interpreted the median score of community pharmacists’ attitude, which was 23 (range 19–26 out of 30), as a positive attitude towards MTM [33]. The study conducted in Malaysia reported the positive attitude of hospital pharmacists based on the average scores for each statement, such as on the role expansion through MTM services (4.48 ± 0.619) and the MTM benefits to patient therapy (4.34 ± 0.651) [36]. The study by Herbert et al., (2006) in the USA reported a mean summary score of 24.16 ± 4.18 for seven questions to interpret pharmacists positive attitude [43].

Akonoghrere et al., (2020) in Nigeria found that pharmacists strongly agreed on their role in MTM (70.3%) and the benefits of MTM on therapy outcomes and the pharmacist—patient relationship (>60%) [34]. Therefore, the authors stated that the pooled attitude toward providing MTM services was found to be extremely positive. The study conducted in Lebanon showed that >70% of community pharmacists were likely to perform medication reviews and counseling [37]. The authors concluded that the pharmacists had a positive attitude regarding MTM.

Another study conducted in Nigeria reported pharmacists’ positive attitudes based on yes/no answers to questions about attitude toward MTM [35]. The results demonstrated that almost all pharmacists agreed that MTM services should be provided, and 75% expressed an interest in MTM training [35]. The willingness of the pharmacists to promote MTM was considered as a positive attitude toward MTM [35]. Law et al., (2009) also evaluated the attitude toward MTM based on pharmacists’ willingness to provide the service [41]. They used agreements on six positive statements concerning self-perceived preparedness and willingness to provide MTM services. According to their study, pharmacists reported having adequate clinical knowledge (95.7%) and experience (90.1%) to provide MTM services, and they believed that they should (99.3%) and were willing to (78.4%) provide MTM services [41]. Hence, Law et al., concluded that community pharmacists reported being ready, willing, and able to provide MTM services [41].

Although other studies have indicated positive attitudes, the study conducted by Shah and Chawla (2011) reported that pharmacists had a slightly positive attitude toward MTM provision [40]. This conclusion was based on the mean summary score of 32.22 ± 3.73 from the agreements on 12 positive statements about attitude with a 5-point Likert scale (5 = strongly agree) [40].

Furthermore, Battaglia et al., (2012) evaluated the training program by measuring attitude based on agreements on four positive statements with a 7-point Likert scale (7 = strongly disagree) [39]. The mean summary scores were 6.6 (pretraining) and 7.2 (post training) (*p* = 0.16) [39]. The authors concluded that the program did not significantly improve attitudes [39].

The last study on attitude was that conducted by MacIntosh et al., (2009) [42]. It evaluated the attitude between pharmacy managers contracted by Mirixa and those not contracted by Mirixa using yes/no answers to 10 questions or positive statements. The results demonstrated that 96% (of contracted by Mirixa) and 88% (of not contracted by Mirixa) believed that they have the qualification to provide MTM (*p* = 0.01) [42]. Moreover, 59% (of contracted by Mirixa) and 45% (of not contracted by Mirixa) strongly agreed that annual PMR would improve therapy outcomes (*p* = 0.04), and both of them (>85%) believed that MTM benefits clinical outcomes [42]. Significantly, more pharmacists contracted by Mirixa believed in their qualification to provide MTM and strongly agreed about MTM benefits to therapy [42].

Regarding the validity and reliability of the questionnaire, five studies reported content validation [32,33,36,37,40], nine studies conducted pre-testing [32,33,34,35,36,37,40,42,43], and four studies reported reliability through Cronbach’s alpha scores that were >0.70 [32,36,40,43]. Two studies conducted by Battaglia et al., (2012) and Law et al., (2009) did not report the instrument validation, including no content validity, pre-testing, or reliability [39,41].

The studies included in our analysis also determined the predictors of attitude or the influence of the attitude itself on other variables. For instance, the study conducted by Akonoghrere et al., (2020) in Nigeria discovered a positive relationship between participant characteristics and attitudes [35]. They found that additional qualification was possibly associated with attitude (F = 5.987, *p* = 0.005) [35]. Another study conducted by Jarab et al., (2022) in Jordan found a positive correlation between knowledge and attitude (r = 0.37, *p* < 0.01) [33]. Law et al., (2009) in the USA also identified that knowledge and practice were significantly associated with attitude (willingness to provide MTM), wherein higher knowledge, competence, and experience were related to a more positive attitude to MTM provision [41]. Furthermore, the study conducted by Herbert et al., (2006) in the USA found that attitude significantly influenced pharmacists’ intention to provide MTM (R^2^ = 0.632; *p* = 0.05) [43].

### 3.4. Pharmacists Practice

A total of 13 studies were conducted on practice toward MTM, which are presented in Table 4. Five studies reported that pharmacists had practiced some MTM elements [33,34,35,40,47], one study reported the challenges for providing MTM [33], whereas all 13 studies reported either perceived or actual barriers to MTM provision [33,34,35,36,37,40,41,42,44,45,46,47,48]. Two studies also determined the facilitator of the provision of MTM services [44,47], and five studies evaluated the training needs that could support MTM implementation [36,37,46,47,48].

Most of the studies used yes/no questions and scales/rates to identify the practice of MTM. The respondents admitted their involvement in some MTM elements. Community pharmacists in Jordan had a moderate level of practice (median: 24, ranging from 21–28 out of 40), with the most frequently provided service being assessments of the health status (84.8%) and the least was services documentation and communication with other providers (62%) [33]. In Nigeria, most community pharmacists claimed to have practiced MTM services [34], whereas hospital pharmacists had practiced almost all MTM elements except MTR [35]. Over 40% of community pharmacists in New York City had performed comprehensive and targeted reviews, but >70% had not yet performed PMR and MAP [40]. Furthermore, 27.1% of community pharmacists in West Virginia reported that MTM services were available in their pharmacies [47].

Jarab et al.’s (2022) study in Jordan evaluated the challenges in providing MTM using a 5-point Likert scale of eleven choices, with 1-point being for ‘strongly disagree’ [33]. The study identified that pharmacists had reported a moderate level of challenges score (median: 33, ranging from 26–45 out of 55) [33]. Collecting patient-related information (36.8%) and providing drug-related information (25.2%) were the most and the least commonly reported challenges, respectively [33]. Other common challenges to providing MTM services in the study included referring the patient to a physician (36.0%) and collaborating with the physician or consultant (35.6%) [33].

All studies reported the barriers to MTM provision, and either used yes/no questions or scales to the questions or statements about barriers, checklists, or multiple-choices. The majority of studies reported insufficient time as the most significant barrier to the provision of services [34,35,36,40,41,42,44,47,48]. Other most frequently reported barriers were insufficient compensation [34,35,40,41,47,48], staff [33,35,37,40,41,45,46], space [37,40,48], and training [33,35,36,48]. One study also identified that negative physician attitudes was the most reported barrier (40.4%) [33]. The study conducted by Lounsbery et al., (2009) in the USA reported that in every case, the barriers were higher for those interested in than for those who already provide MTM services [45]. The most significant actual barriers were related to compensation, and they were greater for uncompensated than for compensated providers [45].

Two additional studies identified the facilitators of MTM provision based on Likert scale scoring, with a high score representing the facilitator. Both studies identified patients’ willingness to participate (5.69 ± 1.17932; 5.81 ± 1.0835) and educational background (5.24 ± 1.49832; 5.65 ± 1.1935) as the facilitators of services [44,47].

In terms of training needs, the study conducted in Malaysia reported that >90% of pharmacists were interested in learning more about the service, and 80% intended to participate in online training [36]. According to the study conducted in Lebanon, >60% of pharmacists required training to become active MTM providers [37]. Moreover, in the study conducted by Bright et al., (2009) in the USA, 74.3% of pharmacists were more likely to provide MTM if they had more training [46]. Another study conducted in the USA by Blake et al., (2009) identified that pharmacists preferred diabetes (3.21 ± 0.95) and hypertension (3.09 ± 0.94) topics in disease-state management training [47]. Interest in additional training was also reported by 78% of pharmacists in the study conducted by Moczygemba et al., (2008) in the USA [48].

Almost all studies on practice (*n* = 12) pretested the survey questionnaire, whereas only six studies performed content validation. Two studies reported instrument reliability, with Cronbach’s alpha values of 0.822 in the study conducted by Al-Tameemi and Sarriff and 0.90 in the confidence section of the study conducted by Moczygemba et al., [36,48].

Several studies attempted to determine the factors that influence pharmacists’ practice toward MTM. The study on Nigerian hospital pharmacists revealed that there was a correlation between age and practice (F = 3.078, *p* = 0.031), educational degree and practice (F = 6.249, *p* = 0.003), and additional qualification and practice (F = 7.383, *p* = 0.002) [35]. Another study on community pharmacists in the USA found that years of practice, pharmacy type, documentation system, and experience in patient care services affected confidence in providing MTM (*p* < 0.05) [48]. Pharmacists with fewer years of practice, more adequate documentation systems, and patient care experience were more confident in providing MTM [48]. Jarab et al.’s (2022) study in Jordan also identified the correlation between challenges scores and both knowledge and attitudes of r = 0.22 and r = 0.32 (*p* < 0.01), respectively [33]. The study also found that type of pharmacy (chain or independent) and income were associated with challenges [33].

### 3.5. Quality of Reporting

According to the reporting quality assessment from the CROSS checklist, eleven studies were categorized as high quality [36,37,38,39,40,42,43,44,45,46,48], while six articles were of moderate quality [32,33,34,35,41,47]. Appendix A shows the proportion of each item in the CROSS checklist either reported sufficiently, partially, or not at all by the included studies. Most studies failed to report information on the entry process, such as how to minimize human error regarding data entry in a paper-based survey or how to prevent multiple submissions in an online survey. The studies also failed to provide details about how to handle missing data. Representativeness and generalizability also became the issues in the studies reporting quality. Although several studies did not thoroughly assess these criteria, most provided discussions about key findings, limitations, and interpretations. Only six studies did not disclose the funding source or potential conflicts of interest.

## 4. Discussion

This study has revealed that among the 17 studies on knowledge, attitude, and practice toward MTM, 5 (29.4%) reported that pharmacists had a considerable level of knowledge, seven (41.8%) suggested a positive attitude among pharmacists, and five (29.4%) discovered that pharmacists had been practicing some elements of MTM. According to the knowledge studies, pharmacists had a moderate or good knowledge level of different concepts of MTM. Some studies defined the knowledge of MTM as knowledge of the definition, core elements, goals, beneficiaries, roles, and providers of MTM [32,33,36], while another defined MTM knowledge as awareness of MTM, pharmaceutical care, and their similarities [35]. A study by Domiati et al., (2018) defined it as familiarity with MTM and knowledge about the process of avoiding drug-related problems [37]. In addition, Al-Tameemi and Sarriff (2019) included some questions about MTAC in the evaluation of MTM knowledge [36]. Although they had different concepts of MTM, a high level of knowledge could indicate a higher likelihood of MTM implementation success than a low level of knowledge [51]. 

A positive attitude reported by most studies could also indicate a higher likelihood of MTM implementation success. Since a negative attitude can be a significant barrier in healthcare services provision [52,53,54,55,56], particularly in MTM service delivery [3], the finding implies a high chance for MTM implementation. Pharmacists may have developed a favorable attitude toward MTM because of their experience with pharmaceutical care, which is the philosophy of MTM [1,35]. Studies reported that pharmacists also realized their expansion role from merely dispensing medications to patient-oriented care, including medication review to prevent or resolve medication errors [33,36].

In terms of practice, only several studies reported pharmacist involvement in the core elements of MTM. The data could provide a more detailed picture of the likelihood of MTM implementation success. This is due to the fact that, even with adequate knowledge and positive attitudes, logistical barriers may limit MTM implementation [51]. Every study identified different elements in the most and least commonly practiced. Thus, every setting had unique characteristics regarding the practice of MTM.

This study discussed not only the KAP level but also the predictors of the variables. It was found that knowledge is related to attitude [33,35,41], and attitude influences practice [33,43,44]. Moreover, experience has been linked to knowledge as well as effects on attitude [36,41]. In terms of knowledge—attitude and attitude—practice correlations, the findings are similar to those of Rahmah et al., (2021) [57]. According to Puspitasari et al., (2020) [58], knowledge has the potential to influence attitudes directly. Consequently, expanding knowledge is critical to fostering a positive attitude and good practice [58].

Other predictors were related to the sociodemographic characteristics of pharmacists. Factors associated with knowledge may include age, educational degree, income, and pharmacy practice setting at the hospital [33,36]. This finding is in line with the study by Biresaw et al., (2020) [59]. These factors may increase the likelihood of becoming more involved in the services [59], which will help pharmacists understand the service better. Regarding attitude, additional qualifications may become its predictor [35]. A possible explanation is that education could influence how people act and behave [59]. Age, educational degree, additional qualification, years of practice, pharmacy type, documentation system, and experience in patient care services could become predictors of MTM practice [33,35,48]. In accordance with Nadew et al., (2020) [60], system-related factors, such as pharmacy type and documentation system, were reasonably associated with the practice.

In addition, understanding the challenge, barriers, and facilitators of MTM provision is essential for successful MTM implementation. The review identified that collecting patient-related information, referring the patient to a physician, and collaborating with the physician or consultant were the most commonly reported challenges [33]. These issues have previously been discussed in reviews by Oladapo and Rascatti (2012) and Ferreri et al., (2020) [3,25]. Patients’ perceptions of pharmacists’ capability in healthcare services might cause the data collection problem. Patients saw pharmacists as merely medication dispensers rather than healthcare professionals who work alongside their doctors [3]. They are content with the traditional dispensing model and do not believe that additional services are necessary [3]. The problem could also be attributed to poor pharmacist-physician collaboration, which resulted from physicians’ negative perceptions of pharmacists’ ability to provide MTM services [25]. The research proposed strategies for improving pharmacist-patient and pharmacist-healthcare team relationships to address these challenges [3,25].

The most reported barriers towards MTM provision were insufficient time [34,35,36,40,41,42,44,47,48], followed by insufficient staff [33,35,37,40,41,45,46], compensation [34,35,40,41,47,48], and training [33,35,36,48]. Similar to the findings in the study by Moore et al., (2017) on person-centered care (PCC), insufficient time-constrained PCC delivery [61]. The rapid pace of healthcare activities made the service implementation difficult. However, once embedded, the service saved time when patients took responsibility for their care [61]. The difficulties with not having enough time are usually associated with a lack of staffing [3]. Studies suggested increasing team membership, particularly trained pharmacy technicians who could play a role in MTM scheduling, billing, patient contact, and documentation, to overcome staffing barriers [3,46]. Regarding compensation, Ferreri et al., (2020) proposed developing a sustainable business model for MTM [3]. MTM must be aligned with value-based care delivery through the development of payment models [3].

Addressing the lack of training barriers could become a vital point. Besides increasing knowledge [38,39], training could improve pharmacist skills, allowing them to overcome other constraints such as time, additional staff, and interprofessional relationships [3,40]. As discussed in other systematic reviews, training is suggested as a method to improve the practice or implementation of services [62,63]. Furthermore, most pharmacists expressed an interest in MTM training. In the United States, Bright et al., discovered that most pharmacists would be more likely to provide MTM if they received more training and would not feel comfortable providing the services without additional training [46]. Domiati et al., in Lebanon reported that 64.5% of pharmacists were willing to attend advanced training sessions to become actively involved in MTM, even if they would not be compensated [37]. The training needs should be adequately fulfilled to improve pharmacists’ competencies as providers of MTM services [25].

The review found that patients’ willingness to participate and educational background were the facilitators of MTM provision [44,47]. Pharmacists who perceive a high value of patient care and feel that they are able and comfortable in responding to patient interest in MTM services are more likely to provide the service [44]. Consistent with our finding, a systematic review by Li et al., (2019) on evidence-based practice (EBP) reported that most facilitators belong to the providers part and relate to the improvement of the ability and values of EBP both in an individual and an organizational context [64]. These findings highlight the importance of advanced practice experiences to build pharmacists’ confidence in providing MTM services [44].

The included studies used different survey delivery methods, including direct (hand-delivered), online, mail, and telephone methods. We also found that the direct delivery method may be associated with a higher response rate than other methods such as the online method that was more preferred recently. This was shown in the studies by Akonoghrere et al., (2020), Al-Tameemi and Sarriff (2019), Domiati et al., (2018), Shah and Chawla (2011), which has a response rate of over 70% [34,35,36,37,40]. This finding is consistent with the statement of Nulty (2008) and Nayak and Narayan (2019) that, in general, online surveys are much less likely to achieve response rates as high as those achieved by hand-delivered surveys [65,66]. The response rate is important in a survey study because it represents the actual sample number that determines the generalizability of the study findings [67]. Therefore, the reader should consider the response rate to take the findings into account. The reader should also be aware of the limitations of survey design, such as low generalizability and self-selection in online surveys. Online surveys are prone to unknown population distributions and sampling biases, resulting in unreliable results [68]. It typically reaches individuals who are literate and sufficiently interested in the topic (respondents select themselves into the sample) [66,68]. In this case, the results of the surveys must be regarded as tentative [68].

Most studies recruited community pharmacists rather than hospital pharmacists. The unique position of community pharmacists to provide healthcare services may cause the choice of this population of interest [69,70]. Because the included studies used different assessment methods, we could not conduct a meta-analysis or pool the data to identify differences in KAP levels between community and hospital pharmacists. The findings reported by Lounsbery et al., (2009) and Al-Tameemi and Sarriff (2019) may provide insights into those hospital pharmacists who were less likely to agree with MTM provision barriers, which were related to interprofessional relationships, documentation, and physical space [36,45].

Methodological aspects may become a concern since most of the studies did not report the results of the methodological quality assessment. Although most studies pretested the questionnaires, only the studies conducted by Alshehri et al., (2022) and Al-Tameemi and Sarriff (2019) performed all validity and reliability tests [32,36], and only five studies reported the value of Cronbach’s alpha [32,36,40,43,48]. The lowest value of Cronbach’s alpha reported was 0.326 for the knowledge section of the survey in Malaysia [36], while the highest value was 0.9 for the confidence section of the survey in the USA [48]. The lowest value reported was far below the lowest limit of a reliable survey, 0.7 [71]. However, unreported or unassessed reliability values make interpretation difficult. The validity and reliability of the questionnaire are critical because they represent the accuracy and precision of the measurement of KAP. Furthermore, most studies did not specify the cut-off value as the basis for KAP-level interpretation. Only the study by Jarab et al., (2022) in Jordan detailed the interpretation mechanism for each KAP level based on the median of the calculated scores [33]. The methodological issues are similar to the review by Manocci et al., (2020) [62]. Thus, this study agreed with the focus on the need for a methodologically high-quality survey [62].

Medication therapy management has already become a widespread practice [72]. Several studies in China, Ethiopia, Germany, Indonesia, South Africa, and Brazil have reported the success of MTM in improving clinical, economic, and humanistic outcomes [6,7,10,12,73,74,75,76]. Additionally, studies in Germany, Thailand, and Sweden have reported positive patient feedback about the services [77,78,79]. However, KAP surveys among pharmacists are still lacking in studies. This could be due to the confidentiality of KAP research in a particular country or to a different approach to program design and development.

A KAP survey is beneficial for the development of an effective MTM program. The KAP level may serve as a predictor of whether pharmacists will actively engage as MTM providers. In addition, information about the variables that may influence the KAP level could be used to design an intervention to improve program effectiveness. This study encourages researchers and policymakers to administer the KAP survey on their own to create a locally relevant and effective MTM program. Findings about challenges, barriers, and facilitators could also be used to improve MTM services provision and ensure its sustainability. Therefore, this study is relevant for countries that will provide or have provided MTM services already.

The first strength of our study lies in the focus on pharmacists as potential MTM providers. We sought a comprehensive point of view from the potential providers of MTM services. Pharmacists’ KAP, including reported challenges, barriers, and facilitators, is essential to develop effective targeted interventions to support MTM program development. Second, we evaluated pharmacists throughout the world. Hence, the study sheds new light on the KAP level and barriers outside the USA. Third, this study considered the validity and reliability of the questionnaires used. Questionnaire validation, such as pilot testing and reliability testing, should be described in the studies. It is also crucial to consider response rate in the KAP study because it represents the actual participant number, thus determining the generalizability. Finally, we discussed the characteristics of the participants because each setting allowed for a different response. As MTM provides a framework, the program implementation can consider the specific demographic aspects and conditions of the disease/patient.

Nonetheless, there were some limitations in this systematic review. First, we only used the PubMed and EBSCO databases. However, both of these databases are relevant and comprehensive on the topic of MTM. We also performed a supplemental search by handsearching the relevant studies. Second, as we only included quantitative surveys, so our findings may differ from those of other quantitative and qualitative KAP studies. Nevertheless, the findings of quantitative surveys may provide an initial depiction and be useful for direct program evaluation. Another limitation was the possibility of publication bias due to the lack of a search for gray literature. We only included peer-reviewed published articles to ensure the comparability of the study quality. Future research must include other quantitative and qualitative KAP studies to obtain a comprehensive approach to MTM provision.

## 5. Conclusions

A total of 17 articles on pharmacists’ KAP toward MTM were examined. Five studies reported that pharmacists had a considerable level of knowledge, seven studies suggested a positive attitude among pharmacists, and five studies found that pharmacists had been practicing some MTM elements. Factors associated with pharmacists’ KAP may include age, educational degree, additional qualification, income, years of practice, practice setting, and experience in patient care service. The challenges toward MTM provision were related to the pharmacist-patient and pharmacist-physician relationship. Insufficient time, staff, compensation, and training were the barriers, while patients’ willingness to participate and educational background were the facilitators of MTM provision. These findings of studies on KAP could help in developing an MTM program and designing an intervention to improve program effectiveness. Researchers and policymakers could administer the KAP survey in their settings to develop a locally relevant and effective MTM program. Further research focusing on other quantitative and qualitative KAP studies is needed to obtain a comprehensive approach to MTM provision.

## Figures and Tables

**Figure 1 healthcare-10-02513-f001:**
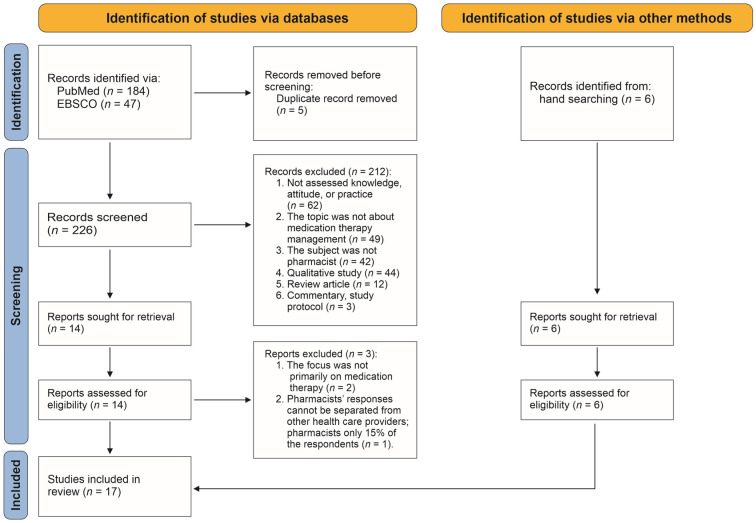
PRISMA flowchart: the study selection.

**Table 1 healthcare-10-02513-t001:** Characteristics of the studies.

Characteristics	Number of Studies
KAP studied ^1^:	
Knowledge	7
Attitude	11
Practice	13
Survey delivery method:	
Mail	4
Telephone	1
Online	7
Direct	5
Participants:	
Community pharmacist	12
Hospital pharmacist	2
Pharmacist in outpatient settings	1
Pharmacist (in general)	2
Country:	
USA	11
Nigeria	2
Jordan	1
Lebanon	1
Malaysia	1
Saudi Arabia	1
Participants number:	
<250	12
250–500	3
>500	2

^1^ A single study could assess all knowledge, attitude, and practice variables.

**Table 2 healthcare-10-02513-t002:** Studies on knowledge.

Reference	Study Design & Instrument	Population	Response Rate (*n*)	Participants’ Characteristics			Assessment Method	Validity and Reliability of Instrument	Main Findings	Authors’ Interpretation	Funding
				Years of Age	Educational Degree	Years of Practice					
Alshehri et al., 2022 [32]	Cross-sectional; Online, self-administered questionnaire	Pharmacists, Saudi Arabia	NR (149)	Mean ± SD: 30.33 ± 6.91	B.Pharm.: 46.31%	Mean ± SD: 4.55 ± 5.43	True/False questions	Content validated, pre-tested; reliability was not reported	Mean ± SD score: 4.30 ± 1.00	Pharmacists had good knowledge about MTM.	Deputyship for Research and Innovation, Ministry of Education in Saudi Arabia
Jarab et al., 2022 [33]	Cross-sectional; Online, self-administered questionnaire	Community pharmacists, Jordan	NR (250)	Median (range): 26 (24–30)	B.Pharm.: 67.6%	1–5: 73.6%	Multiple-choice and True/False questions	Content validated, pre-tested; reliability was not reported	Median score: 6 (range 4–7 out of 10). Age and income positively associated with higher knowledge score (*p* < 0.01); lower educational degree associated with less knowledge score (*p* < 0.01).	The pharmacists showed a moderate level of knowledge towards MTM.	None
Akonoghrere et al., 2020 [35]	Cross-sectional; Direct, self-administered questionnaire	Hospital pharmacists, Nigeria	86.96% (100)	31–40: 34% 41–50: 34%	B.Pharm.: 76%	<10: 65%	Yes/No and multiple-choice questions	Pre-tested; reliability was not reported	94% knew MTM. 98% admit the similarities between MTM and pharmaceutical care.	Most pharmacists had good knowledge of MTM.	NR
Al-Tameemi and Sarriff, 2019 [36]	Cross-sectional; Direct, self-administered questionnaire	Hospital pharmacists, Malaysia	71.5% (93)	Mean ± SD: 29.06 ± 5.14	B.Pharm.: 88.2%	0–10: 90.3%	Multiple-choice and True/False questions	Content validated, pre-tested; Cronbach’s α = 0.326	Median score = 9 (range 6–10 out of 10). There was a difference in knowledge level in the practice setting (*p* = 0.004).	Most pharmacists had a high knowledge level of MTM.	None
Domiati et al., 2018 [37]	Cross-sectional; Direct, surveyor-administered questionnaire	Community pharmacists, Lebanon	82% (820)	Mean ± SD: 40.41 ± 11.17	B.Pharm.: 49.9%	≥10: 59.5%	Familiarity level and Yes/No questions	Content validated, pre-tested; reliability was not reported	46% were familiar with the MTM concept. More than 60% of those willing to provide the service were knowledgeable of avoiding drug-related problems.	The community pharmacists have adequate knowledge towards MTM implementation.	None
Brown et al., 2018 [38]	Longitudinal; Online, self-administered training assessment	Community pharmacists and pharmacy technicians, USA	57.14% (16)	NR	NR	Mean (range): 7 (0.2–15.5)	Multiple-choice questions with confidence level of answer correctness	NR	Mean correct answer (%) adjusted by participants’ confidence: pretraining = 71.0, posttraining = 82.4 (*p* < 0.05)	The training was an effective and essential method to ensure the pharmacy staff skills of MTM.	The North Dakota Department of Health and The North Dakota Pharmacy Services Corporation.
Battaglia et al., 2012 [39]	Longitudinal; Online, self-administered training assessment	Pharmacists, USA	34.43% (42)	Mean ± SD: 38.4 ± 9.72	NR	NR	Multiple-choice questions	NR	Mean correct answers: pretraining- = 7.1, posttraining = 8.0, change = 0.8 (*p* < 0.01)	The program improved participants’ knowledge of diabetes-MTM.	National Association of Chain Drug Stores

NR = not reported, SD = standard deviation, B.Pharm. = Bachelor of Pharmacy, MTM = medication therapy management.

**Table 3 healthcare-10-02513-t003:** Studies on attitudes.

**Reference**	**Study Design & Instrument**	**Population**	**Response Rate (*n*)**	**Participants’ Characteristics**			**Assessment Method**	**Validity and Reliability of Instrument**	**Main Findings**	**Authors’** **Interpretation**	**Funding**
				**Years of Age**	**Educational Degree**	**Years of Practice**					
Alshehri et al., 2022 [32]	Cross-sectional; Online, self-administered questionnaire	Pharmacists, Saudi Arabia	NR (149)	Mean ± SD: 30.33 ± 6.91	B.Pharm.: 46.31%	Mean ± SD: 4.55 ± 5.43	Degree of importance scale, with 7 = beneficial/good/pleasant for me/useful	Content validated, pre-tested; Cronbach’s α = 0.850	Mean ± SD score: 6.15 ± 1.12	The pharmacists had a positive attitude towards MTM services.	Deputyship for Research and Innovation, Ministry of Education in Saudi Arabia
Jarab et al., 2022 [33]	Cross-sectional; Online, self-administered questionnaire	Community pharmacists, Jordan	NR (250)	Median (range): 26 (24–30)	B.Pharm.: 67.6%	1–5: 73.6%	Agreements on positive statements with scales: 5 = strongly agree	Content validated, pre-tested; reliability was not reported	Median score: 23 (range 19–26 out of 30). Attitudes- knowledge correlation (r = 0.37, *p* < 0.01).	Pharmacists showed a positive attitude towards MTM services.	None
Akonoghrere et al., 2020 [34]	Cross-sectional; Direct, self-administered questionnaire	Community pharmacists, Nigeria	94% (118)	31–40: 39.8% 20–30: 35.6%	B.Pharm.: 70.3%	5–10: 35.6% 1–5: 20.3%	Agreements on positive statements	Pre-tested; reliability was not reported	Pharmacists strongly agreed on their role in MTM (70.3%) and the MTM benefits on the therapy outcomes and pharmacist-patient relationship (>60%).	The pooled pharmacists’ attitude toward providing MTM was very positive.	NR
Akonoghrere et al., 2020 [35]	Cross-sectional; Direct, self-administered questionnaire	Hospital pharmacists, Nigeria	86.96% (100)	31–40: 34% 41–50: 34%	B.Pharm.: 76%	<10: 65%	Yes/No to positive questions	Pre-tested; reliability was not reported	97% agreed that MTM should be provided. 75% interested in MTM training. Additional qualification-attitude correlation (F = 5.987, *p* = 0.005).	Pharmacists were willing to promote MTM, indicating a positive attitude.	NR
Al-Tameemi and Sarriff, 2019 [36]	Cross-sectional; Direct, self-administered questionnaire	Hospital pharmacists, Malaysia	71.5% (93)	Mean ± SD: 29.06 ± 5.14	B.Pharm.: 88.2%	0–10: 90.3%	Agreements on positive statements with scales: 5 = strongly agree	Content validated, pre-tested; Cronbach’s α = 0.716	Average score on: pharmacists’ advanced role (4.46 ± 0.635) and the role expansion through MTM services (4.48 ± 0.619).	Most pharmacists had a positive attitude towards MTM provision.	None
Domiati et al., 2018 [37]	Cross-sectional; Direct, surveyor-administered questionnaire	Community pharmacists, Lebanon	82% (820)	Mean ± SD: 40.41 ± 11.17	B.Pharm.: 49.9%	≥10: 59.5%	Agreements on positive and negative statements	Content validated, pre-tested; reliability was not reported	Over 70% were likely to perform medication reviews and counselling.	Most pharmacists had a positive attitude regarding MTM.	None
Battaglia et al., 2012 [39]	Longitudinal; Online, self-administered training assessment	Pharmacists, USA	34.43% (42)	Mean ± SD: 38.4 ± 9.72	NR	NR	Agreements on positive statements with scales: 1 = strongly agree	NR	Mean summary scores: pretraining = 6.6, posttraining = 7.2, change = 0.6, (*p* = 0.16)	The program did not improve attitudes.	National Association of Chain Drug Stores
Shah and Chawla, 2011 [40]	Cross-sectional; Direct, self-administered, questionnaire	Community pharmacists, USA	73.81% (93)	Mean ± SD: 48.08 ± 11.80	NR	Mean ± SD: 20.04 ± 11.63	Agreements on positive statements with scales: 5 = strongly agree	Content validated; Cronbach’s α = 0.750	Mean ± SD summary scores: 32.22 ± 3.73.	Pharmacists had a slightly positive attitude toward MTM provision.	None
Law et al., 2009 [41]	Cross-sectional; Online, self-administered questionnaire	Community pharmacists, USA	1.43% (143)	51–60: 34.3% 41–50: 31.5%	NR	NR	Agreements on positive statements	NR	Pharmacists believed they should (99.3%) and were willing to (78.4%) provide MTM. Knowledge and experience were strongly related to attitude.	Community pharmacists reported being ready, willing, and able to provide MTM services.	NR
MacIntosh et al., 2009 [42]	Cross-sectional; Telephoned, surveyor-administered questionnaire	Independent pharmacy managers, USA	20% (200)	NR	NR	NR	Yes/No to questions/positive statements	Pre-tested; reliability was not reported	59% (of contracted) and 45% (of not contracted) strongly agreed to the benefit of annual PMR (*p* = 0.04).	More pharmacists contracted with Mirixa strongly agreed that an annual PMR would improve patient outcomes.	Mirixa
Herbert et al., 2006 [43]	Cross-sectional; Mailed, self-administered, questionnaire	Community pharmacists, USA	41% (203)	NR	B.Pharm.: 75.9%	21–30: 29.1% 1–10: 28.6%	Agreements on statements with scales: 5 = strongly agree (reverse coded for negative statements)	Pre-tested; Cronbach’s α = 0.812	Mean ± SD summary scores: 24.16 ± 4.18 Attitude influenced intention to provide (R^2^ = 0.632; *p* = 0.05).	Pharmacists generally showed a positive attitude in providing MTM.	NR

NR = not reported, SD = standard deviation, B.Pharm. = Bachelor of Pharmacy, MTM = medication therapy management, PMR = personal medication record.

**Table 4 healthcare-10-02513-t004:** Studies on practice.

Reference	Study Design & Instrument	Population	Response Rate (*n*)	Participants’ Characteristics			Assessment Method	Validity and Reliability of Instrument	Main Findings	Authors’ Interpretation	Funding
				Years of Age	Educational Degree	Years of Practice					
Jarab et al., 2022 [33]	Cross-sectional; Online, self-administered questionnaire	Community pharmacists, Jordan	NR (250)	Median (range): 26 (24–30)	B.Pharm.: 67.6%	1–5: 73.6%	Responses on statements with scale: 5 = always	Content validated, pre-tested; reliability was not reported	Median practice score: 24 (range 21–28 out of 40). Median challenges score: 33 (range 26–45 out of 55). Challenges—knowledge correlation (r = 0.22, *p* < 0.01). Challenges—attitudes correlation (r = 0.32, *p* < 0.01). Type of pharmacy and income were associated with challenges. The most reported barriers: negative physician attitudes (40.4%), lack of training (38.4%), and lack of adequate support staff (37.2%).	The pharmacists had performed or obtained necessary assessments of the patient’s health status.	None
Akonoghrere et al., 2020 [34]	Cross-sectional; Direct, self-administered questionnaire	Community pharmacists, Nigeria	94% (118)	31–40: 39.8% 20–30: 35.6%	B.Pharm.: 70.3%	5–10: 35.6% 1–5: 20.3%	Yes/No/Don’t know questions	Pre-tested; reliability was not reported	93.3% admitted providing MTM services. Concerns about MTM provision: job satisfaction, time, and compensation.	MTM services were closely associated with the practice among pharmacists surveyed.	NR
Akonoghrere et al., 2020 [35]	Cross-sectional; Direct, self-administered questionnaire	Hospital pharmacists, Nigeria	86.96% (100)	31–40: 34% 41–50: 34%	B.Pharm.: 76%	<10: 65%	Yes/No questions	Pre-tested; reliability was not reported	Pharmacists had been practicing almost all core elements of MTM, except MTR. The most significant barriers were insufficient staff, time, and training. There was a correlation between age and practice (F = 3.078), educational degree and practice (F = 6.249), qualification and practice (F = 7.383); *p* < 0.05.	Pharmacists have started providing some of MTM the service.	NR
Al-Tameemi and Sarriff, 2019 [36]	Cross-sectional; Direct, self-administered questionnaire	Hospital pharmacists, Malaysia	71.5% (93)	Mean ± SD: 29.06 ± 5.14	B.Pharm.: 88.2%	0–10: 90.3%	Yes/No questions (“No” scored 0, “Yes” scored 1)	Content validated, pre-tested; Cronbach’s α = 0.822	Over 90% were interested in providing and learning more about MTM. The barriers identified were inadequate training (88.2%), funds (51.6%), and time (46.2%).	Most pharmacists were eager to learn more about the service. The most perceived barriers to service provision were inadequate training.	None
Domiati et al., 2018 [37]	Cross-sectional; Direct, surveyor-administered questionnaire	Community pharmacists, Lebanon	82% (820)	Mean ± SD: 40.41 ± 11.17	B.Pharm.: 49.9%	≥10: 59.5%	Agreement on positive statement, Yes/No, and multiple-choice questions	Content validated, pre-tested; reliability was not reported	More than 60% expressed an interest in advanced training to become active providers. Less than 50% had a private consultation area, as well as adequate workflow, time and staff.	Pharmacists were interested in advanced training. However, the barriers identified were insufficient time, workflow, and space.	None
Shah and Chawla, 2011 [40]	Cross-sectional; Direct, self-administered, questionnaire	Community pharmacists, USA	73.81% (93)	Mean ± SD: 48.08 ± 11.80	NR	Mean ± SD: 20.04 ± 11.63	Yes/No and rating on questions	Content validated; reliability was not reported	Over 40% had provided comprehensive and targeted medication reviews; more than 70% had not yet provided PMR and MAP. Barriers: 80% admitted additional pharmacists, time, and compensation, while 70% admitted technical staff, space, and documentation.	Pharmacists have been involved in some elements of MTM. Additional time and staff would be the most difficult MTM challenges.	None
Blake and Madhavan, 2010 [44]	Cross-sectional; Mailed, self-administered questionnaire	Community pharmacists, USA	29.2% (256)	NR	Pharm.D.: 18.9%	Mean ± SD: 18.9 ± 12.4 (1–53)	Rating: 7 = strongly agree; high scores = facilitators, low scores = barriers.	Content validated, pre-tested; reliability was not reported	The lowest scores: insufficient time (3.08 ± 1.878); the highest scores: patients’ willingness to participate (5.69 ± 1.179), educational background (5.24 ± 1.498). The likelihood to provide MTM was influenced by comfort level and ability.	Pharmacists indicated that insufficient time was the barrier, while patients’ willingness to participate and pharmacists’ education were the facilitators to the provision of MTM.	None
Law et al., 2009 [41]	Cross-sectional; Online, self-administered questionnaire	Community pharmacists, USA	1.43% (143)	51–60: 34.3% 41–50: 31.5%	NR	NR	Agreements on positive statements	NR	The most frequent responses to the MTM barriers: “Each health plan has a different specification of MTM services”, “Lack of time and staffing”, and “Uncertainty of reimbursement”.	Despite being ready, willing, and able to provide, pharmacists need assistance in the MTM process.	NR
MacIntosh et al., 2009 [42]	Cross-sectional; Telephoned, surveyor-administered questionnaire	Independent pharmacy managers, USA	20% (200)	NR	NR	NR	Yes/No to questions/positive statements	Pre-tested; reliability was not reported	64% admitted insufficient time to provide patient care services, and 65% admitted difficulty finding time to meet with patients one-on-one.	Despite the existing barriers, contracting and understanding the requirements increases the likelihood to provide MTM.	Mirixa
Lounsbery et al., 2009 [45]	Cross-sectional; Online, self-administered questionnaire	Pharmacists in the outpatient settings, USA	6.7% (970)	Mean ± SD: 40.2 ± 11.9	NR	Mean ± SD: 15.1 ± 12.2	Agreements on questions with scales: 5 = strongly agree	Pre-tested; reliability was not reported	Perceived barriers: insufficient staff (89.6%), access to medical information (84%). Actual barriers (compensated vs. uncompensated): insufficient compensation (70.8%, 75.7) and billing knowledge (62.2%, 81.1%), inability to obtain compensation (67.3%, 83.2%).	The perceived barriers were higher than the actual barriers. The most significant barriers were related to compensation and were greater for uncompensated than for compensated providers.	American Pharmacists Association Foundation Incentive Grant for Practitioner Innovation in Pharmaceutical Care
Bright et al., 2009 [46]	Cross-sectional; Mailed, self-administered questionnaire	Community pharmacists, USA	29.2% (256)	NR	B.Pharm.:68%	<5: 30.2% >20: 22.9%	Yes/No questions	Pre-tested; reliability was not reported	Training topics preference: MTM elements (75.3%), systems (62.5%). Intention to provide MTM: if had more training (74.3%), if had more staff (77.8%). 75% need assistance in at least scheduling and billing,	Technician assistance may help to reduce pharmacist staffing and training needs.	None
Blake et al., 2009 [47]	Cross-sectional; Mailed, self-administered questionnaire	Community pharmacists, USA	40.4% (203)	NR	NR	NR	Responses on question and statements with Likert scale (low scores = barriers, and high scores = facilitators)	Content validated; pre-tested; reliability was not reported	27.1% reported that MTM services were available in their pharmacies. Facilitators: patients’ willingness to participate (5.81 ± 1.08), educational background (5.65 ± 1.19). Barriers: insufficient time (2.88 ± 1.82), reimbursement (3.79 ± 1.95).	Pharmacists were providing MTM services in their pharmacies. They were comfortable providing it but reported that insufficient time was the barrier.	NR
Moczygemba et al., 2008 [48]	Cross-sectional; Online, self-administered questionnaire	Community pharmacists, USA	11.8% (157)	Mean ± SD: ±9.8	NR	Mean ± SD: 25.7 ± 10.8	Responses on questions and statements with 4 or 5-point scale	Pre-tested; Cronbach’s α = 0.9 in confidence section	Mean ± SD confidence scores of conducting MTM elements (4 = very confident): overall MTR = 3.0 ± 0.6, create a PMR = 3.3 ± 0.8, makes referral = 3.2 ± 0.7. Mean ± SD barriers (4 = very challenging): time (3.2 ± 0.8), compensation (3.2 ± 0.8), physical environment. Fewer years of practice, more adequate documentation, and patient care experience, more confident.	Pharmacists were eager to expand their roles as patient care providers. Their confidence suggested that they already had many necessary skills but required training to become successful providers.	NR

NR = not reported, SD = standard deviation, B.Pharm. = Bachelor of Pharmacy, Pharm.D. = Doctor of Pharmacy, MTM = medication therapy management, MTR = medication therapy review, PMR = personal medication record, MAP = medication action plan.

## Data Availability

The data presented in this study are available in the manuscript and Appendix A.

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
