# Peer review of "Pharmacists’ Knowledge, Attitude, and Practice of Medication Therapy Management: A Systematic Review"

_healthcare, 2022, doi:10.3390/healthcare10122513_

Round 1

Reviewer 1 Report

This is a well-written report that sets out how the systematic review was carried out and sets out the findings. 

A little more information about the spread of MTM outside of the US would be helpful.   You may have searched for papers globally, but you find few countries where research that met your inclusion criteria had been carried out. There are more than 180 countries that are members of the UN.

You do not say in your inclusion criteria if you included papers in languages other than English. 

You discuss the limitations of online surveys. However, I think you could also have considered the issue of self-selection. It is possible that pharmacists with knowledge of / who use MTM services disproportionately self-select to respond to the questionnaire. Online surveys are also less likely to have a probability sample than survey-administered ones.

You may also like to consider the difference between significance, unlikely to have been due to chance, and large enough difference to make it worthwhile too, e.g. give pharmacists more training.  

A minor point you say that one article reported a CA of 0.326. I thought that 0.7 was the lower limit for a CA?

Reviewer 2 Report

Overall: Rendrayani et al conducted a systematic review to assess pharmacists’ knowledge, attitude, and practice of MTM. Overall, the review is well written. My comments are minor:

Abstract

1.       Line 17: what is the starting date of the search?

2.       What are the keywords for searching?

3.       Which measurement tools for MTM KAP would the authors recommend for future studies?

Introduction

1.       I like this review is focusing the MTM around the world, however, given the difference in healthcare systems (the practicing scope of pharmacists varies by countries, US most requires PharmD degree while other countries have BPharm), it will be helpful to introduce the differences of MTM in different countries. And maybe compared them with US MTM.

Methods

3.1   search strategy: authors only mentioned starting date of the search, but missed the end date of searching

Results

1.       Line 247: why this paper reported a “slightly” positive attitude? What are the cut-offs for a positive attitude?

Discussion

1.       Line 379-380: it’s better to include the percentages after the number of papers too.

2.       As mentioned in the abstract, since different assessment methods were used for different studies, I think it will be helpful to add recommendations for measurement tools/questionnaires to use for future studies.
